# Polyphenol Intake in Pregnant Women on Gestational Diabetes Risk and Neurodevelopmental Disorders in Offspring: A Systematic Review

**DOI:** 10.3390/nu14183753

**Published:** 2022-09-11

**Authors:** Blanca Salinas-Roca, Laura Rubió-Piqué, Anna Montull-López

**Affiliations:** 1Global Research on Wellbeing (GRoW) Research Group, Blanquerna School of Health Science, Ramon Llull University, Padilla, 326-332, 08025 Barcelona, Spain; 2Department of Nursing and Physiotherapy, University of Lleida, Montserrat Roig 2, 25198 Lleida, Spain; 3Antioxidants Research Group, Food Technology Department, AGROTECNIO-CERCA Center, University of Lleida, Av/Alcalde Rovira Roure 191, 25198 Lleida, Spain

**Keywords:** pregnant women, polyphenol, gestational diabetes, fetal neurodevelopment, transgenerational health

## Abstract

The intake of foods containing polyphenols can have a protective role to avoid comorbidities during pregnancy and, at the same time, promote transgenerational health. This review aims to describe the effect of polyphenol intake through supplements or polyphenol-rich foods during pregnancy on the incidence and evolution of gestational diabetes mellitus (GDM), as well as the link with the neurodevelopment of the fetus. Using PRISMA procedures, a systematic review was conducted by searching in biomedical databases (PubMed, Cinahl and Scopus) from January to June 2022. Full articles were screened (*n* = 419) and critically appraised. Fourteen studies were selected and were divided into two different thematic blocks considering (i) the effect of polyphenols in GDM and (ii) the effect of GDM to mental disorders in the offspring. A positive relationship was observed between the intake of polyphenols and the prevention and control of cardiometabolic complications during pregnancy, such as GDM, which could be related to thwarted inflammatory and oxidative processes, as well as neuronal factors. GDM is related to a greater risk of suffering from diseases related to neurodevelopment, such as attention deficit hyperactivity disorder, autism spectrum disorder and learning disorder. Further clinical research on the molecule protective mechanism of polyphenols on pregnant women is required to understand the transgenerational impact on fetal neurodevelopment.

## 1. Introduction

Transgenerational health describes the impact of exposures prior to conception and during gestation on the health of offspring [1]. Diet and nutritional status of the mother before and during pregnancy can determine both the health of the fetus and her own. During pregnancy, the mother’s body must satisfy the growing physiological needs that this process entails [2,3,4], as well as during breastfeeding [2], to achieve a correct fetal development [4].

The placenta has three main functions: the exchange of gases and nutrients between the fetus and the pregnant woman, the elimination of fetal waste and the production of enzymes and hormones [5]. Nutrients are transmitted to the placenta according to the size and charge of the molecules available to be transported, the concentration of nutrients in the fetal and maternal blood, and the lipid solubility of the particles that are transported [6]. Small molecules that have little or no charge, such as water, or lipids, such as cholesterol, can cross the placenta more easily [7]. Similarly, the absorption of polyphenols through the placenta is believed to involve selective transporter mechanisms, suggesting that these compounds are able to cross the placental barrier and, therefore, may have biological effects on the offspring [8]. Larger molecules, such as enzymes and insulin, cannot cross it [6,8]. The inability of insulin to cross the placenta leads to higher blood glucose levels, since glucose can cross the placenta, and, therefore, the fetus receives more energy than it needs [7].

During pregnancy, metabolic changes can occur that favor insulin resistance [6,9]. The nutritional needs of pregnant women are aimed at providing energy to the fetus [9]. Mainly, carbohydrates (monosaccharides) and fats are the nutrients responsible for the energy supply [9]. When changes in hormonal levels occur, the pregnant woman’s body tends to use lipids to make glucose available to the fetus, and this increases the amount of free fatty acids, which alter insulin sensitivity [10]. The increase in insulin resistance in pregnant women leads to a higher level of glucose in the fetal circulation when crossing the placenta [11]. Thus, pregnant women can suffer from different cardiometabolic complications, such as gestational diabetes mellitus (GDM), obesity or hypertensive pathologies.

In Europe the prevalence of GDM is 10.9% [12]. Women with GDM are at greater risk of requiring an induced or caesarean delivery, of suffering from preeclampsia or of developing hypertension [3]. Malnutrition on the part of the mother can have negative effects on her own health and on that of the baby; thus, maintaining a healthy diet can help in the prevention of complications [2,3,4]. Therefore, nutrition plays an important role in GDM during gestation, as well in children’s obesity prevalence [13]. Moreover, mental disorders in children have been increased in the recent years. Recent studies have been relating dietetic habits as a cause of mental illness in offspring [14].

Healthy eating recommendations include the intake of 500 g of fruit and vegetables [15,16]. This group of foods contains micronutrients, such as polyphenols, which perform antioxidant functions. Polyphenols are natural components found in vegetables, fruits, cereals and certain drinks. These are divided into [17] phenolic acids (including benzoic acids, which are mostly present in tea leaves and red fruits, and cinnamic acids, which are mostly found in practically all fruits and grain cereals); flavonoids (including flavanols, flavonols, anthocyanins, flavones, isoflavones and flavanones) found in most fruits, green and black tea, chocolate, onions, broccoli, parsley, cilantro and soy; stilbenes, present mainly in red wine and red grape juice; lignans, found in flax and sesame seeds; and others such as turmeric, chlorogenic acid, rutin and silibinin, which are mostly present in most fruits, coffee, turmeric and thistle seeds.

The biological properties of polyphenols are extensive: They have vasoprotective and vasodilating effects, as well as antithrombotic effects [18]. They also have antilipemic and antiatherosclerotic properties [19]. In the scientific community, these micronutrients have attracted attention given the recent evidence of their role in the prevention of diseases such as cancer, diabetes mellitus, cardiovascular diseases and neurodegenerative pathologies [17]. Thus, the different effects of polyphenols on health can lead to both benefits and consequences for fetal health due to the different properties of these components, including their antioxidant and anti-inflammatory effects [17,18,19,20].

The use of foods containing polyphenols can have a beneficial effect by helping to avoid comorbidities during pregnancy and, at the same time, promote transgenerational health. An inverse association has been determined between the consumption of foods rich in polyphenols and the risk of chronic non-communicable diseases (NCDs) in adults. On the other hand, research carried out in pregnant women is very scarce [21].

In recent years, it has been shown that nutritional recommendations delivered together with the recommendations for lifestyle changes are effective in preventing pregnancy complications and improving maternal metabolism [22,23,24]. Nevertheless, the role of polyphenol intake on maternal and fetus health should be addressed more in the literature. Thus, this review aims to describe the effect of polyphenol intake during pregnancy on the incidence and evolution of gestational diabetes mellitus, as well as the link with the neurodevelopment of the fetus.

## 2. Materials and Methods

### 2.1. Research Question

The research question is as follows: what effect can the intake of polyphenols exert on gestational diabetes mellitus risk and the good neural development of the fetus?

### 2.2. Identification of Relevant Studies

For the identification of studies, articles were searched in the databases PubMed, Cinahl Plus and Scopus until March 2021. With reference to search strategies, for the first search, the keywords used were “polyphenols”, “polyphenol intake”, “gestational diabetes” and “pregnancy diabetes”. Search combinations varied across databases. In both PubMed and Cinahl Plus, the combination was ((polyphenols) OR (polyphenol intake)) AND ((gestational diabetes) OR (pregnancy diabetes)). In Scopus, the combination was “polyphenol intake” AND (“gestational diabetes” OR “pregnancy diabetes”). For the second search, the keywords “gestational diabetes”, “neurodevelopmental disorders” and “foetus” were used. In all databases the combination was (gestational diabetes) AND (neurodevelopmental disorders) AND (foetus).

### 2.3. Inclusion and Exclusion Criteria

The inclusion criteria for the selection of studies were original research performed in humans, including clinical trials, case–control, longitudinal cohort, and cross-sectional and case report studies. The studies using polyphenol-rich foods, as well as polyphenol supplements, were taken into consideration for the systematic review. Articles from more than 10 years, duplicates, books and revisions or meta-analysis, as well as works written in non-English language, were excluded.

### 2.4. Selection of Relevant Studies

The review was conducted in accordance with the PRISMA guideline [25]. The PRISMA flow diagram of the total bibliographic search shows the classification of the articles resulting in a total of 433 articles. However, the 14 studies selected in the qualitative and quantitative synthesis are divided into two different tables, one for each search. In these, the following information is included: Table 1 contains the first author and reference, country, year of publication, study design, sample size (*n*), sample characteristics, method and intervention period, polyphenol dosage, outcomes measured and main results on the risk of GDM. Table 2 also includes the first author and reference, country, year of publication, type of study, sample size (*n*), sample characteristics, data taken into account and main results. The methodological quality of the included studies was assessed by using the adapted CASPe guide, where all studies presented a total score between 7 and 9 out of 9.

## 3. Results

The total number of articles resulting from the search is 433, from which 14 were included according to the eligibility criteria (Figure 1). Of these 14 studies, 7 (50% of the total studies) are cohort studies, 2 are case-control studies (14.28%), 2 are prospective longitudinal studies (14.28%), 1 is a randomized parallel arm study (7.14%), and 2 are prospective randomized double-blinded placebo-controlled clinical trials (14.28%).

The results obtained in the present review are grouped into two different thematic blocks, considering (i) the effect of polyphenols in GDM (Table 1), where 4 articles were found, and (ii) the effect of GDM to mental disorders in the offspring (Table 2), with 10 articles.

Table 1 presents articles related to polyphenol intake and cardiometabolic complications in pregnant women. The results obtained included both polyphenol-rich foods (blueberries) and polyphenol supplements (resveratrol and epigallocatechin 3-gallate). Malvasi et al. carried out a study with three groups of overweight pregnant women, one supplemented with 80 mg of trans-resveratrol (Revifast^®^), together with D-chiro-inositol/Myo-inositol (DCI/MI); another only with DCI/MI; and a placebo group, with the aim of investigating the effect of trans-resveratrol and DCI/MI on lipid profile and glucose levels [26]. On the other hand, the study by Basu et al. investigated the effect of daily supplementation with blueberries rich in anthocyanins (700 mg/day) with soluble fiber on the cardiometabolic profile of mothers at risk of GDM and the control group [27]. Similarly, Gao et al. studied the association of polyphenols in the diet and the risk of GDM by comparing the consumption of these natural components in a total of 2231 pregnant women [28]. Finally, Zhang et al. determined in 472 pregnant women a beneficial effect of epigallocatechin 3-gallate supplementation (500 mg/day) on maternal diabetic parameters and pregnancy and fetus complications [29].

Table 2 shows the articles (*n* = 10) in which the effects of GDM on the risk of suffering from diseases related to neurodevelopment are described. Krakowiak et al. conducted a study on this topic and declared the association between GDM and mental disorders affecting language in children [30]. Nomura et al. examined the effect of GDM together with a low socioeconomic status (SES) on attention deficit hyperactivity disorder (ADHD) and neurodevelopment in 212 preschoolers [31]. In the study by Zornoza-Moreno et al., the temperature maturation of the circadian rhythm, activity and sleep in the first year of life of 63 infants of mothers with diabetes mellitus (DM) was analyzed [32]. Girchenko et al. examined the association between gestational obesity and developmental delay and whether this is caused by or given worse outcomes by DM or hypertension (HT) in pregnancy in 4785 Finnish children [33]. In the study by Despina et al., it was determined whether the expression of brain-derived neurotrophic factor, nerve growth factor and neurotrophin-4 are different in children who are large for gestational age, children with intrauterine growth restriction or children with appropriate growth for gestational age depending on whether the mother is diabetic or not [34]. Torres-Espínola et al. studied the effect of maternal pathologies such as overweight, obesity and GDM on the visual evoked potentials of a total of 331 children [35]. In the study by Kong et al., the combined effects of maternal obesity and DM on the development of psychiatric pathologies and mild neurodevelopmental disorders were determined in a total sample of 649,043 infants [36]. Moreover, Panjwani et al. studied the association of maternal metabolic conditions with branched-chain amino acids (BCAA) and the risk of ASD in 789 mother–child pairs [37]. Akaltun et al. investigated the relationship of ADHD and learning disorder (LD) with diabetic pregnancy in 265 women [38]. Finally, Morgan et al. studied the effect of biomarkers such as glycated hemoglobin (Hb1Ac), *C*-reactive protein and blood pressure on the executive function of 100 infants [39].

Dietary polyphenols are well-known for their potential health benefits, including their role in improving glycemic control and in managing obesity. This review assesses the human evidence regarding the consumption of polyphenols in relation to the risk of GDM. According to the results presented in Table 1, a positive relationship is observed between the intake of polyphenols and the prevention and control of cardiometabolic complications during pregnancy such as GDM. In the study by Basu et al., through the supplementation of blueberries, a decrease in the concentration of glucose in pregnant women and an increase in gestational weight were observed [27]. In addition, there is an inverse correlation between serum glutathione and the oxidative capacity of PAI-1. According to Gao et al., the total intake of polyphenols and polyphenols from fruit is associated with a lower risk of GDM, specifically the intake of flavonoids, as this was inversely associated with the incidence of GDM [28]. Regarding the interventions providing polyphenol supplements, in the study by Malvasi et al. [26], an improvement or decrease in blood glucose levels and lipid profile was observed in the group with trans-resveratrol supplementation. In the study performed by Zhang et al. [29], it was reported that daily epigallocatechin 3-gallate supplementation in safe doses (500 mg/day) is able to improve maternal diabetic symptoms and neonatal outcomes (low birth weight and hypoglycemia) of GDM-affected women. Since some of the dietary interventions involved a combination of blueberries and soluble fiber [27] and trans-resveratrol with DCI/MI [26], identifying the individual effects of the studied polyphenols cannot be performed accurately in these studies.

According to pharmacokinetic studies in rat maternal plasma and fetuses, the maternal plasma concentrations of catechins were ~10 times higher than in placenta and 50–100 times higher than in the fetus [40]. At several time points after administration, blood samples were collected, and placental and fetal tissues were obtained. Moreover, Arola-Arnal et al. (2013) [40] reported that flavanols and their metabolites were widely distributed in both pregnant and non-pregnant rats’ plasma and tissues. Conjugated forms of flavanols were more abundant in the livers of non-pregnant rats compared with pregnant rats, suggesting that flavanol metabolism is less active during pregnancy. Furthermore, flavanol metabolites were abundant in the placenta and detected at low levels in the fetus and amniotic fluid. Overall, this suggests that these compounds are able to cross the placental barrier and, therefore, may have biological effects on the offspring.

The findings of our review corroborated prior research regarding the beneficial role of adherence to a Mediterranean diet rich in polyphenol-rich-food in the reduced risk of GDM [41]. In this meta-analysis, it was shown that the risk of GDM was about halved amongst women with the highest score of Mediterranean diet compared to those with the lowest score. These results are also consistent with another meta-analysis on adherence to a Mediterranean diet and type 2 diabetes [42], which shares some pathophysiological similarities with GDM.

The results also agree with the antidiabetic effect of polyphenols described by Kim et al. [43] that points to a relationship of these natural components with the inhibition of enzymes such as salivary and pancreatic α-amylase and α-glucosidase and the inhibition of glucose absorption. Insulin resistance is characterized by a lower response to insulin in the appropriate tissues, and, consequently, the β-cells responsible for generating it continue to do so, and a global malfunction occurs due to oxidative stress due to the state of hyperglycemia, hyperinsulinemia and the inflammatory factor [44].

Insulin resistance is the result of a continuous generation of reactive oxygen species (ROS) due to constant inflammation and/or high glucose concentrations in the tissues. These can unbalance the metabolism and cause a greater production of pro-inflammatory cells. The lack of insulin action means that even more is produced due to the overactivity of β cells, which also have low levels of antioxidant substances, and overall oxidative stress is facilitated [44]. Moreover, α-amylases and α-glucosidases are responsible for digesting starch to produce glucose, which then passes into the blood. The digestion of starch and sugar leads to a glucose spike and, consequently, a rapid response of the β-cells to produce insulin, and the formation of ROS occurs. However, some polyphenols can inhibit α-amylases and α-glucosidases, alleviating the glucose peak and lower ROS concentrations [43,44].

According to Martel et al. [45], the different types of polyphenols play a major role in the transport of various bioactive components both in the placenta and in the intestinal epithelium. The development and function of the placenta are altered by the presence of GDM. The increase in the total syncytiotrophoblast surface and greater vascularization of the placental villi lead to an increase in the fetal–placental endothelial surface. Glucose transport is achieved through facilitated diffusion with GLUT glucotransporters. This transport is regulated according to the expression of GLUTs, the concentration of extracellular glucose and the absorption and release of this [46].

Polyphenols are secondary metabolites and the main contributors to the antioxidant action of fruits, even more so than vitamin C. This is possible thanks to the fact that they can donate an electron or hydrogen atom, neutralizing free radicals [47]. In addition, they reduce the rate of oxidation by preventing the formation of active oxygen species or deactivating them. Another ability of polyphenols is the induction of antioxidant enzymes such as catalase, glutathione and superoxide dismutase.

Alternatively, polyphenols have the ability to suppress inflammation and thus oxidative damage by disrupting oxidative-stress-signaling pathways and abrogating the signaling transducer mechanism of proinflammatory mediators [48]. Vegetables and fruits with a high content of polyphenols, such as apples, citrus fruits, artichokes, pears, fruits of the forest, plums, onions, endives and broad beans, among others, are considered neuroprotective, being able to modulate cellular processes such as the signaling, proliferation and differentiation of the redox balance or apoptosis [47,49]. Thus, the anti-inflammatory and antioxidant action of polyphenols has the capacity to regulate the decompensation of processes such as oxidative stress.

In pregnant women with GDM or pregestational diabetes, increased placental levels of GLUT1, GLUT 4 and GLUT 9 have been observed. However, the increase of this GLUT1 in a hyperglycemic state can lead to an increase in fetal glucose levels, and consequently it can stimulate the response of pancreatic β cells before time, enabling the promotion of fetal overgrowth, and even in situations without hyperglycemia, it can lead to higher levels of glucose transport [50]. An alternative to be considered is the modulation of the microbiota in the GDM. Probiotics are microorganisms that promote health benefits to the host. Bifidobacterium and Lactobacillus are the most widely used for this purpose. This procedure can promote the better composition of the intestinal microbiota; can reduce the adherence of pathobionts; can strengthen intestinal permeability; can aid the immune response, insulin signaling and energy metabolism; can be a safe alternative; is well-tolerated; and has proven beneficial effects in various clinical conditions, including GDM [51].

Aside from the reported beneficial effects of polyphenols, some detrimental effects are attributable to their pro-oxidant activities when high doses are administered, generating free radicals under certain conditions. All the revised studies in the present review stated that the daily polyphenol doses are within the safety limit of consumption. Moreover, current studies have suggested that maternal ingestion of polyphenol-rich food, especially during the third trimester of pregnancy, could be associated with fetal ductal constriction (DC) [52]. The deleterious effects of polyphenols, such as DC, in late pregnancy might be restricted to this special condition and population, as the reports on beneficial effects still outnumber those on the eventual negative effects.

GDM is related to a greater risk of suffering from diseases related to neurodevelopment and behavior, such as attention deficit hyperactivity disorder, autism spectrum disorder and learning disorder [30,31,36,37,38]. However, according to Girchenko et al., children of mothers with GDM are more likely to suffer a minimal delay in the development of social skills, as well as not being able to develop normal communication skills for the child’s age, nor develop properly the thick engine [33]. In their cohort study, Despina et al. observed a deficit of brain-derived neurotrophin and nerve growth factor in children born to mothers with GDM with intrauterine growth restriction, associated with sequelae in neurodevelopment and compromise of cortical function, as well as neurocognitive deficits and a higher incidence of later psychiatric pathologies [34]. Otherwise, in the article by Morgan et al., no effect of HbA1c on cognitive flexibility or effects of pro-inflammatory factors on the child’s cognitive response was observed [39].

In the review by Rowland et al., neurodevelopmental impairment and GDM are related to oxidative stress, which, at the same time, can lead to behavioral problems due to motor deficits [53]. Oxidative stress is an imbalance of antioxidant and pro-oxidant substances, with an excess of the latter. This imbalance leads to a greater reduction in cellular potential, as well as a lower reducing capacity of cellular redox couples, such as glutathione [53]. In the case of patients with diabetes mellitus, the concentration of antioxidants such as vitamin A and E, albumin and transferrin, among others, decreases. This increase in free radicals as a result of greater peroxidation can lead to damage to organs such as the kidney and the central nervous system; and molecules such as DNA, proteins and lipids [54,55]. High blood glucose levels from long exposure led to an overproduction of ROS. This excess and/or underutilization of endogenous antioxidants can cause oxidative stress, causing β-cell dysfunction and insulin resistance by disrupting insulin signaling with its receptor to activate absorption of glucose [56].

GDM can lead to global DNA hypermethylation or altered RNA expressions. High levels of this methylation have been associated with a repression of gene expression; however, it may vary according to the location of the methylation in the gene sequence [57]. In a study of a sample of 817 children, 13 genomic locations were identified with DNA methylation levels that can predict ADHD symptoms. Although the literature is scarce in regard to this topic, evidence describes that, in children with ASD, GDM is associated with greater deficits in language expression, albeit slightly [30]. Developmental delays are observed and are more common in mothers with diabetes, obesity or hypertension. However, Kong et al. [36] associates pregestational diabetes mellitus with a risk >6 times greater than in a non-diabetic or healthy-weight mother but does not observe an increased risk in the case of women with GDM. Glutathione is one of the most important antioxidants in the prevention of cell damage due to excess reactive oxygen species. Thus, the brain is the organ that uses the most oxygen, and, consequently, it is the most vulnerable to oxidative stress. Children are even more affected by having a lower concentration of antioxidants, in addition to excessive ROS and mitochondrial dysfunction, all characteristics that have the potential to cause oxidative stress [31,32,38,58].

Certainly, in the study by Zornoza Moreno et al. in the group of children of mothers with GDM, a greater gestational age was highlighted, as well as a greater abdominal circumference and caesarean-section rate. In this, a greater perimeter is correlated with the values that indicate a worse maturation of the circadian rhythm; in addition, a better regulation of this rhythm is determined in the control group than in the group with GDM + insulin [32]. The circadian rhythm is a set of brain structures and peripheral organs that allows the signal of external stimuli to be transported to the circadian clock, located in the suprachiasmatic nucleus of the hypothalamus [59]. On the other hand, in the study by Torres-Espínola et al. in the children of women with GDM, remarkably poor visual acuity was observed at three months of age, as well as prolonged latencies in visual evoked potentials at 18 months of age, suggesting a negative effect of this disease in myelination and postnatal brain development [35].

Moreover, the educational dimension is related to a better diagnosis and treatment of pregnant women’s complications [60,61,62]. According to the review of Espuig Sebastián et al., in a study conducted on 23 pregnant women in Sweden, although they received oral and written information, they ended up looking for information on the Internet due to confusion or fear to talk about their diet [63]. However, in a study of 15 pregnant women in Norway, little information was found. Pregnant women agreed that the information was very general and that, despite being giving information in writing, the reinforcement of this oral information was minimal [63]. In the same review, a lack of education on the part of the midwives was observed, as well as a lack of security on the part of these professionals to give the information.

It is worth mentioning that this review presents several limitations. In the methodological field, the bibliographic search in the different databases resulted in a large number of total articles, even if the vast majority of these does not fit the keywords searched. The keyword combinations were modified several times to achieve a more accurate search for the chosen topic. In addition, the vast majority of articles that did fit the keywords and inclusion and exclusion criteria turned out to be reviews or meta-analyses. Research on the effect of polyphenols on GDM, in particular, and on the action of this disease on fetal neurodevelopment and epigenetics is scarce. More clinical evidence is required to confirm the anti-diabetic effect of certain polyphenols or polyphenol-rich foods on the development of GDM in pregnant women. Moreover, this work has shown the potential therapeutic effect of certain polyphenols on GDM during pregnancy, but the safety of their long-term use needs to be deeply studied. It is also important to develop exhaustive safety studies before recommendations can be made in this at-risk population. Not all antioxidants of natural origin can be considered suitable for this population, as certain antioxidants have shown teratogenic effects in mice [64].

## 4. Conclusions

In summary, a positive relation between polyphenols and blood glucose levels is described in the literature. Moreover, GDM is the cardiometabolic complication that is most related to various pathologies and consequences in fetal neurodevelopment. The period of pregnancy is crucial for giving the pregnant woman the necessary information to maintain healthy habits for herself and the fetus.

The existing relationship between GDM and a higher incidence of neurodevelopmental diseases, such as ADHD, TA and ASD, is highlighted, as well as other brain development problems, such as a worse development of visual evoked potentials and difficulties in the correct regulation of circadian rhythms.

The antioxidant and anti-inflammatory actions of polyphenols can prevent harmful processes in the body, such as oxidative stress, and, therefore, they can also prevent its consequences, such as metabolic decompensations, various diseases (e.g., GDM) and changes in DNA as is its methylation. In summary, polyphenols have a positive impact on transgenerational mother–child health and, therefore, on epigenetics. Therefore, novel therapies should be more investigated on the modulation of the intestinal microbiota, with probiotics and prebiotics, and the use of natural products with antioxidant and anti-inflammatory properties, which could mitigate the endogenous processes of cardiometabolic disorders in pregnant women.

## Figures and Tables

**Figure 1 nutrients-14-03753-f001:**
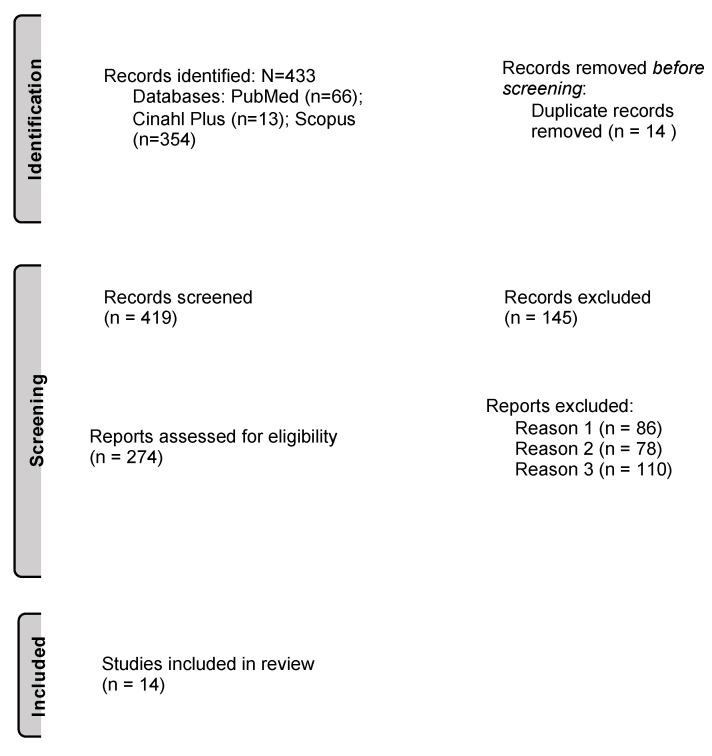
Flow diagram showing the search, identification and selection process for the reviewed articles and the reports excluded because of the following reasons: the work (1) does not discuss polyphenols’ action and gestational diabetes mellitus, (2) does not discuss gestational diabetes mellitus in relation to neurodevelopment and the fetus and (3) is a review. Adapted with permission from Page et al. (2021) [25].

**Table 1 nutrients-14-03753-t001:** Characteristics of studies in the effect of polyphenol intake regarding the possibility of developing GDM.

Study and Country	Year	Study Design	*n*	Pregnant Women Characteristic	Pregnancy Period	Study Groups and Period of Assessment	Polyphenol Dosage	Outcomes Measured	Main Findings on GDM Risk
Malvasi et al. [26], Italy	2017	Prospective randomized double-blinded placebo-controlled clinical trial	110	Pregnant and overweight women between 25 and 40 years old, first trimester’s gestational BMI between 25 and 30 kg/m^2^.	Between 24th and 28th pregnancy week	Group 1: supplemented with trans-resveratrol in combination with myo-inositol and D-chiro-inositol. Group 2: only receives myo-inositol and D-chiro-inositol. Group 3: control group (placebo). Sixty days.	Group 1: 80 mg of trans-resveratrol, 200 mg of myo-inositol, 500 mg of D-chiro-inositol. Group 2: 138 mg of myo-inositol, 550 mg of D-chiro-inositol.	SBP, DBP, Total cholesterol, LDL, HDL, triglycerides and blood-glucose level before and after the first 30 and 60 days of treatment.	No difference between systolic and diastolic parameters of the three groups. Blood parameters improved compared to placebo group after 30 days, and even more after 60 days compared with Groups 1 and 2. Group 1 presented larger improvement in the lipid profile and blood-glucose levels than Group 2.
Basu et al. [27], USA	2021	Randomized parallel arm study	34	Obesity (BMI ≥ 30 kg/m^2^), gestational age < 20 weeks with risk of GDM, singleton pregnancy, not having pregestational chronic diseases.	Between <20th and 32/36th pregnancy week	Group 1: under intervention with blueberries (two cups a day) and 12 g of soluble fiber. Group 2: under standard prenatal control. Eighteen weeks.	Two cups of l blueberries, 1600 mg of total polyphenols and 700 mg of anthocyanins.	Gain of gestational weight, blood-glucose levels and *C*-reactive protein in blood. Antioxidant biomarkers in maternal serum, adipokines in serum and hormonal biomarkers. Trace elements in plasma/blood? With a role in antioxidant/oxidative stress pathways (Se, Fe, Zn, Mg and Cu).	↓ of blood-glucose levels and *C*-reactive protein. ↑ of gestational weight. Inverse correlation of maternal glutathione and the antioxidant capacity with PAI−1 (first r: −0.32 and −0.28, respectively; in the end, r: −0.30 and −0.33).
Gao et al. [28], China	2021	Prospective cohort study	2231	Pregnant women between 18 and 45 years old.	Between the 8th and 16th pregnancy week.	FFQ of 61 items, asking about the food frequency and portion during 4 weeks. Forty-one months.	An average of 319.9 mg of total polyphenols (201.6 mg from fruits)	Maternal clinical and sociodemographic data. Quartiles of diary polyphenols intake (Q1, <226.9 mg/d; Q2, 227–317.9 mg/d; Q3, 318–415.8 mg/d; and Q4, ≥415.9 mg/d), origin of the polyphenols (total polyphenols from fruits and vegetables).	Total intake of polyphenols (specifically from fruits) is associated with a lower risk of GDM. The intake of flavonoids was significantly and inversely associated with the incidence of GDM. *ORs* in the biggest quartile of total flavonoids from fruit was between 0.57 (0.32, 0.99) and 0.58 (0.34, 0.99).
Zhang et al. [29], China	2017	Double-blind randomized controlled trial	404	Pregnant women with singleton pregnancy, between 25 and 34 years old, with a diagnosis of GDM.	From the beginning of the third trimester to term	Group 1: instructed to consume one capsule of 500 mg of EGCG daily. Group 2: instructed to consume one capsule of 500 mg of starch powder as placebo. Three months.	500 mg of ECGC	Maternal clinical and body weight data. Glucose and insulin metabolism: insulin, QUICKI, HOMA-IR and HOMA-β. Neonatal complications at birth: low birth weight, hypoglycemia, respiratory distress, macrosomia, 1 min APGAR, 5 min APGAR.	Circulating glucose and insulin response and metabolism are improved by the intervention with EGCG. Of the 57 patients who could not complete the trial because of incontrollable blood-glucose levels, only 16 were from the EGCG group. EGCG group was also associated with less neonatal complications.

DBP, diastolic blood pressure; EGCG, epigallocatechin 3-gallate; FFQ, Food Frequency Questionnaire; HDL, high-density lipoprotein; HOMA-IR, homeostasis model of assessment of insulin resistance; HOMA-β, homeostasis model of assessment of beta cell function; LDL, low-density lipoprotein; QUICKI, Quantitative Insulin Check Index; SBP, systolic blood pressure.

**Table 2 nutrients-14-03753-t002:** Characteristics of studies in relation with GDM, fetus and neurodevelopmental disorders.

Study, Country	Publication Year	Study Design	*n*	Sample Characteristics	Data Taken into Consideration for the Analysis	Main Findings towards GDM
Krakowiak et al. [30], USA.	2012	Population based, case-control study	1004	Children between 2 and 5 years old. Three groups: control, one with ASD and the other with DD.	Maternal clinical data including BMI, HBP, having any type of diabetes or taking any antidiabetic medication. Child’s gender and age, having any metabolic or neurologic disorder, control center.	ASD and DD are more common in mothers with diabetes, obesity or HBP than in mothers from control group. Diabetes is associated to higher but milder deficits in language expression in the case of children with ASD.
Nomura et al. [31], USA.	2012	Ongoing cohort study	212	Children between 3 and 4 years old, with risk of developing ADHD who went to kindergarten in New York. English-speaking parents. Two groups: “in risk” (>6 symptoms according to their parents) and “typically developing” (<3 symptoms according to their parents).	Maternal, paternal and child clinical data, gender, ethnicity, low birth and socioeconomic status. Child neuropsychological functioning, temperament, and behavioral/emotional functioning in the follow-up (6 years old).	GDM and low SES have a negative impact in the symptoms and diagnose of ADHD. The level of risk between GDM and ADHD changes significantly depending on the family SES. In children exposed to both GDM and low SES, the damage of a wide variety of behavioral functions has been proved. ↑ risk of ADHD, significantly in case of the child being exposed to both GDM and low SES.
Zornoza-Moreno et al. [32], Spain	2013	Longitudinal prospective study	63	Pregnant women between 24 and 28 weeks of gestation, singleton pregnancy, age from 18 to 40 years old, non-smoking and not consuming docosahexaenoic acid supplements during pregnancy. Sample divided in 3 groups: control group, diagnosed with GDM and treated with diet and diagnosed with GDM treated with insulin.	Skin temperature, children activity and sleep time index at 15 days old, 1, 3 and 6 months old. Circadian-rhythm maturation. Gestational age, abdominal circumference at the beginning of the study and at birth, weight, length, BMI, waist/hip ratio at birth and at 3, 6 and 12 months old.	Greater gestational age in newborns from diabetic mothers. ↑ abdominal circumference and cesarean rate in relation to control group. A larger circumference is related to a worse circadian-rhythm maturation. Temperature proves a minor relative amplitude in the group of GDM + insulin at 6 months. Better regulation of the circadian rhythm in control group compared to GDM + insulin group.
Girchenko et al. [33], Finland	2018	PREDO study: prospective pregnancy cohort study	4785	Singleton live-born children between 2006 and 2010 in hospitals of the Southern and Eastern Finland.	Maternal clinical data. Overweight, obesity, pregestational/gestational/ chronic HBP, GDM or Type 1 diabetes mellitus.	Children of mothers with GDM had higher odds of having a mild developmental delay of social skills and of failing to meet the development on communication skills which are typical for the age. Higher odds of not meeting the development on the gross motor and problem-solving skills which are usual for the age.
Briana et al. [34], Greece	2018	Cohort study	60	Pregnant women, in the third trimester of pregnancy, and their children. Different groups: 3 groups of mothers with GDM, one for large for gestational age, the second intrauterine-growth-restricted, and the third appropriate for gestational age. Group four also appropriate for gestational age, but without GDM (control).	Child: birthweight, gestational age, customized centile and gender. Mother: age, parity, delivery mode, GDM. Concentrations of: BDNF, NGF and NT-4.	↓ concentrations of BDNF in fetuses exposed to GDM. Intrauterine-growth-restricted children born to mothers with GDM presented NGF deficiency, which can be associated with neurodevelopmental sequelae, neurocognitive deficits, increasing the incidence for later psychiatric disorders and compromising cortical dysfunction.
Torres-Espínola et al. [35], Spain	2018	PREOBE study: prospective mother–child cohort study	331	Pregnant women with singleton pregnancies and age between 18 and 45 years old, between 12 to 20 weeks of pregnancy. Four groups: healthy weight, overweight, obesity and GDM.	Child: gestational age, gender, anthropometrics and cord-blood laboratory status. Maternal clinical data, marital status and intelligence quotient. Blood-glucose and iron levels.	Children born to GDM mothers had a significantly poorer visual acuity at 3 months and prolonged latencies of visual evoked potentials at 18 months of age. More notorious effects on children of mothers with GDM and obesity. Results suggest GDM has a negative impact on the myelination and in postnatal brain development.
Kong et al. [36], Finland	2018	Large prospective population-based cohort study	649.043	Pregnancies between 2004 and 2014 in Finland. Differentiation between non-diabetic mothers, mothers with pregestational diabetes and mothers with GDM.	Child: Offspring year of birth, gender, any perinatal problem, number of fetuses, mode of delivery (vaginal, instrumental, caesarean and others), Mother: Maternal age, parity, marital status, country of birth, smoking habit, psychiatric disorders, diagnoses of systemic inflammatory disorders and BMI.	Pregestational GDM together with maternal obesity is associated with an increased risk of various pediatric psychiatric and mild neurodevelopmental disorders. These mothers have the risk of delivering a child with ASD > 6 times higher than a non-diabetic mother or a mother with healthy weight. In case of GDM, no increased risk of any type of disorder was observed.
Panjwani et al. [37], USA	2019	Prospective longitudinal and intergenerational cohort study	789 couples (mother–child)	Pregnant women from 2004 to 2015, with metabolic measurements and children attended in *Boston Medical Center* without any development disorder.	Mother clinical data. Child: gender, gestational age and birth weight. Concentration of BCAA.	A lower concentration of BCAA than usual + mother’s obesity or diabetes not related to a larger risk of ASD when compared to mothers without GDM. In higher concentrations, the risk of having a child with ASD is the double.
Akaltun et al. [38], Turkey	2019	Case–control study	265	Children born between 2005 and 2010 in Atatürk to diabetic and non-diabetic mothers in terms of ADHD and SLD.	Gender, age, having ADHD or SLD, mother’s age, child’s intelligence quotient and HbA1c.	In children with diabetic mothers, a lower intelligence quotient compared to the control group has been observed. Larger number of cases of ADHD and SLD than in control group. More present in mothers with insulin-dependent diabetes than in group control.
Morgan et al. [39], USA.	2020	Prospective longitudinal study	100	Children between 4 and 6 years old.	Child data and birth or neonatal-health complications. Cognitive flexibility and response inhibition. Maternal clinical data, level of pro-inflammatory factors (HbA1c, *C*-reactive protein and blood pressure) in pre-pregnancy and in the second and third trimester.	HbA1c and blood pressure do not have any effect in cognitive flexibility. Any of the three pro-inflammatory factors have any effect on the children’s response inhibition.

ADHD, attention deficit hyperactivity disorder; ASD, autism spectrum disorder; BCAA, branch-chained amino acid; BMI, body mass index; DD, development delay; HBP, high blood pressure; BDNF, brain-derived neurotrophic factor; NGF, nerve growth factor; NT-4, neurotrophin-4; SES, socioeconomic status; SLD, specific learning disorder.

## Data Availability

Not applicable.

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
