# Peer review of "Polyphenol Intake in Pregnant Women on Gestational Diabetes Risk and Neurodevelopmental Disorders in Offspring: A Systematic Review"

_nutrients, 2022, doi:10.3390/nu14183753_

Round 1

Reviewer 1 Report

Reviewer comments

Thank you for the opportunity to review this work. In this work, Salinas-Roca et al. conducted a two-part systematic review that aimed to describe the effect of polyphenol intake during pregnancy on the incidence and evolution of gestational diabetes mellitus (GDM), as well as the link with the neurodevelopment of the fetus. Please, find below my comments for your response.

Abstract

The authors should state which databases were used and to which date the article extraction was carried out.

Which time of the pregnancy was the polyphenol taken? Would that include polyphenol supplements or only whole diet sources of polyphenols? The authors should kindly provide clarity on that.  Kindly, indicate the duration of intake?

Introduction

Line 36: Please, put the references by each statement rather than lumping them all together

Line 37-39: Kindly, indicate the reference

Several statements that have been made do not have references supporting them. The authors should please correct that. Statements should have references supporting them.

Materials and Methods

Line 108: The authors should have rather used “polyphenol*”. The use of “polyphenol*” implies that studies that used “polyphenol” or “polyphenols” in the title would be retrieved.

Line 114: The authors should indicate if the consumption of polyphenols in the form of capsule was added to the “inclusion criteria”. What if the polyphenol was obtained from the consumption of polyphenol-rich foods such as cocoa beverage or chocolate or tea for example?

Line 123-126: The authors should indicate the dosage of polyphenol consumed.

The authors could have also considered assessing the quality of the articles selected for the review.

Why did the authors decide to focus on articles published in the last 10 years? I think the total number of articles arrived at didn’t necessitate that especially considering that only 14 papers were finally selected for the two objectives investigated.

Results

Line 129: The authors should state if the review was conducted in accordance with the PRISMA guideline. If yes, they should use the Page et al. (2021) reference to guide that (Page, M.J.; McKenzie, J.E.; Bossuyt, P.M.; Boutron, I.; Hoffmann, T.C.; Mulrow, C.D.; Shamseer, L.; Tetzlaff, J.M.; Akl, E.A.; Brennan, S.E.; et al. The PRISMA 2020 statement: An updated guideline for reporting systematic reviews. BMJ 2021, 372, n71).

The authors should revise the PRISMA flow chart. This review is a two-fold one but with two different objectives. The authors should therefore separate the results by stating clearly how many articles were obtained from the search on “polyphenol and its effect on GDM in pregnant women” and how many were obtained from “GDM and neurodevelopmental challenges of foetus”.

Also, in the exclusion criteria of the PRISMA guideline, the authors should indicate how many articles were excluded on the basis of “Do not discuss about polyphenols’ action and GDM”, “Do not discuss about GDM in relation with ND and fetus” and “Reviews”. They can do that by using “n=…….”.

This review lacks depth in the presentation of the results. For example, the authors should have indicated the exact dosage of polyphenols that were consumed by the pregnant mothers. This is important especially as the authors didn’t indicate if the studies selected were focused on “capsule” intake or “whole diet” intake. More importantly, excess intake of polyphenols can result in the polyphenols undergoing pro-oxidation which may make them lose their antioxidant effects.

Table 1. In the Basu et al study, the authors present fibre in the Table. This is very confusing as the objective of the work was on.

The authors should kindly revise Table 1 to clearly state the dosages and also separate the articles that address “polyphenol and its effect on GDM in pregnant women” and those that address “GDM and neurodevelopmental challenges of foetus”.

Conclusion

The authors should indicate how many articles addressed each of the objectives the conclusion addressed.

Author Response

Manuscript ID: nutrients-1885393

29th August 2022

Reviewer 1 comments

Thank you for the opportunity to review this work. In this work, Salinas-Roca et al. conducted a two-part systematic review that aimed to describe the effect of polyphenol intake during pregnancy on the incidence and evolution of gestational diabetes mellitus (GDM), as well as the link with the neurodevelopment of the fetus. Please, find below my comments for your response.

Thank you for your comments. We appreciate the time and effort that you have dedicated to providing your feedback. We believe that the reviewer comments have identified important issues which required improvement, and that after completion of the suggested edits the revised manuscript has benefitted from an improvement.

Abstract

The authors should state which databases were used and to which date the article extraction was carried out.

As requested by the reviewer, this information has been added to the abstract as follows: “Using PRISMA procedures, a systematic review was conducted by searching in biomedical databases (PubMed, Cinahl and Scopus) from January- June 2022.”

Which time of the pregnancy was the polyphenol taken? Would that include polyphenol supplements or only whole diet sources of polyphenols? The authors should kindly provide clarity on that.  Kindly, indicate the duration of intake?

As suggested by the reviewer, authors have indicated in the abstract that the review was focused on both isolated polyphenol sources (supplements) and also polyphenol-rich food sources.

Given that the abstract is limited to 200 words, the other requested information by the reviewer (duration of the intake, time of the pregnancy) could not be included in the abstract, as it was very variable among different studies. According to the reviewer suggestion, new columns have been included in Table 1 presenting this relevant information in a clearer way.

Introduction

Line 36: Please, put the references by each statement rather than lumping them all together

Following the reviewer suggestion, the references supporting each statement have been included.

Line 37-39: Kindly, indicate the reference.Several statements that have been made do not have references supporting them. The authors should please correct that. Statements should have references supporting them.

As the reviewer 1 suggested, new references throughout introduction have been introduced in those statements that were missing (Lines 40, 43, 44 of the revised manuscript).

Materials and Methods

Line 108: The authors should have rather used “polyphenol*”. The use of “polyphenol*” implies that studies that used “polyphenol” or “polyphenols” in the title would be retrieved.

The title used “polyphenol intake” since the search included also this keyword. Nevertheless, the systematic review also considered the group of polyphenols implying all the chemical family.

Line 114: The authors should indicate if the consumption of polyphenols in the form of capsule was added to the “inclusion criteria”. What if the polyphenol was obtained from the consumption of polyphenol-rich foods such as cocoa beverage or chocolate or tea for example?

As requested by the reviewer, we specified in the revised manuscript the following information in the inclusion criteria: “The studies using polyphenol-rich foods as well as polyphenol supplements were taken in consideration for the systematic review” (Lines 118-119).

Line 123-126: The authors should indicate the dosage of polyphenol consumed.

The proposal of Reviewer 1 has been taken in consideration. Thus, dosage of polyphenol consumed in each study has been added in Table 1 as a new column.

The authors could have also considered assessing the quality of the articles selected for the review.

The methodological quality of the included studies was assessed using the adapted CASPe guide, where all studies presented a total score between 7 and 9 out of 9.

Why did the authors decide to focus on articles published in the last 10 years? I think the total number of articles arrived at didn’t necessitate that especially considering that only 14 papers were finally selected for the two objectives investigated.

Authors considered to extend the period of the articles search, however, no differences on significant interventions have been found changing the focus in the last 10 or 20 years. In fact, the use of polyphenol for pregnant women publications are basically described in the last decade.

Results

Line 129: The authors should state if the review was conducted in accordance with the PRISMA guideline. If yes, they should use the Page et al. (2021) reference to guide that (Page, M.J.; McKenzie, J.E.; Bossuyt, P.M.; Boutron, I.; Hoffmann, T.C.; Mulrow, C.D.; Shamseer, L.; Tetzlaff, J.M.; Akl, E.A.; Brennan, S.E.; et al. The PRISMA 2020 statement: An updated guideline for reporting systematic reviews. BMJ 2021, 372, n71).

According to Reviewer’s 1 suggestion, the reference of Page et al. (2021) has been included in Line 123 of the revised manuscript.

The authors should revise the PRISMA flow chart. This review is a two-fold one but with two different objectives. The authors should therefore separate the results by stating clearly how many articles were obtained from the search on “polyphenol and its effect on GDM in pregnant women” and how many were obtained from “GDM and neurodevelopmental challenges of foetus”. Also, in the exclusion criteria of the PRISMA guideline, the authors should indicate how many articles were excluded on the basis of “Do not discuss about polyphenols’ action and GDM”, “Do not discuss about GDM in relation with ND and fetus” and “Reviews”. They can do that by using “n=…….”.

In agreement with the comment, the information has been clarified in the Results’ section where the following text was added: “The results obtained in the present review are present into two different thematic blocks considering i) the effect of polyphenols in GDM (table 1) where 4 articles were found and ii) the effect of GDM to mental disorders in the offspring (table 2) with 10 articles.”

This review lacks depth in the presentation of the results. For example, the authors should have indicated the exact dosage of polyphenols that were consumed by the pregnant mothers. This is important especially as the authors didn’t indicate if the studies selected were focused on “capsule” intake or “whole diet” intake.

As suggested by the reviewer, the exact polyphenol dosage has been included in Table 1 in a clearer way, and also the results section has been extended. Moreover, a depth revision of Table 1 has been made to clarify the detailed information given for each study.

More importantly, excess intake of polyphenols can result in the polyphenols undergoing pro-oxidation which may make them lose their antioxidant effects.

Considering the reviewer comment, a discussion part on this issue has been included (Lines 93-101). Also in Line 174, the safety issue on polyphenols intake during pregnancy was already discussed.

Table 1. In the Basu et al study, the authors present fibre in the Table. This is very confusing as the objective of the work was on.

According to the comment, details of Basu et al. study have been discussed in the text considering that this dietary intervention involved a combination of blueberries and soluble fiber. And identifying the individual effects of the studied polyphenols cannot be done accurately in this study.

The authors should kindly revise Table 1 to clearly state the dosages and also separate the articles that address “polyphenol and its effect on GDM in pregnant women” and those that address “GDM and neurodevelopmental challenges of foetus”.

As requested by the reviewer, Table 1 has been revised to clearly state the dosage and other relevant information. As shown in the manuscript, two separated Tables are used to present results.

The authors should indicate how many articles addressed each of the objectives the conclusion addressed.

As requested by the reviewer, this information has been included in the results section: “The results obtained in the present review are present into two different thematic blocks considering i) the effect of polyphenols in GDM (Table 1) where 4 articles were found and ii) the effect of GDM to mental disorders in the offspring (Table 2) with 10 articles.”

Reviewer 2 Report

Evaluation of the manuscript by Blanca Salinas-Roca entitled “Polyphenol intake in pregnant women on gestational diabetes risk and neurodevelopmental disorders in offspring: a systematic review” sent to Nutrients.

Line 22: which could be related to thwarted inflammatory and oxidative processes, as well as on neuronal factors. What do you think? The word “thwarted” or another similar in meaning could point at beneficial effects of polyphenolic compounds.

Abstract: More clinical evidence on the effect of polyphenols on GDM and on the action of this disease on fetal neurodevelopment is required. Such sentences in conclusion may suggest that the review is not complete. Could you be more specific about what research area (what specific disorders) should be deeper penetrate?

Style of citation in the text seems strange a bit to me. Instead of “Malvasi A et al, carried out” I would find better “Malvasi et al. [25].

The results and discussion are acceptable BUT in my opinion there is lack of some reports dealt with the risk of overconsumption of polyphenols. Is it possible to add some data? It is known that polyphenols are “double-edge” biomolecules.

Author Response

Manuscript ID: nutrients-1885393

29th August 2022

Reviewer 2 comments

Evaluation of the manuscript by Blanca Salinas-Roca entitled “Polyphenol intake in pregnant women on gestational diabetes risk and neurodevelopmental disorders in offspring: a systematic review” sent to Nutrients.

Thank you for your comments. We appreciate the time and effort that you have dedicated to providing your feedback. We believe that the reviewer comments have identified important issues which required improvement, and that after completion of the suggested edits the revised manuscript has benefitted from an improvement.

Line 22: which could be related to thwarted inflammatory and oxidative processes, as well as on neuronal factors. What do you think? The word “thwarted” or another similar in meaning could point at beneficial effects of polyphenolic compounds.

We appreciate the reviewer comment and agree with this statement. According to the reviewer suggestion, this sentence has been modified in the abstract.

Abstract: More clinical evidence on the effect of polyphenols on GDM and on the action of this disease on fetal neurodevelopment is required. Such sentences in conclusion may suggest that the review is not complete. Could you be more specific about what research area (what specific disorders) should be deeper penetrate?

In agreement with your comment, conclusion in abstract has been changed to: “Further clinical research on the molecule protective mechanism of polyphenols on pregnant women is required to understand transgenerational impact on fetal neurodevelopment”.

Style of citation in the text seems strange a bit to me. Instead of “Malvasi A et al, carried out” I would find better “Malvasi et al. [25].

According to your comment the references have been changed.

The results and discussion are acceptable BUT in my opinion there is lack of some reports dealt with the risk of overconsumption of polyphenols. Is it possible to add some data? It is known that polyphenols are “double-edge” biomolecules.

Considering the reviewer comment, a discussion part on this issue has been included (Lines 93-101). Also in Line 174, the safety issue on polyphenols intake during pregnancy was already discussed.

Round 2

Reviewer 1 Report

Reviewer comments

Thank you for addressing most of the comments I raised. The quality of the manuscript has improved subsequently. Please, find below my minor comments for your revision.

Abstract

Line 19: what is the “nf?”

Introduction

Line 37: please remove (3) from the sentence

Result

Line 141: The authors should revise “present” to “grouped”

Figure 1. The authors should revise the PRISMA flow chart. I suggested that in my earlier comments using the Page et al. (2021) reference. The authors should show the “line arrows”. in the PRISMA flow chart. They should use the Page et al. (2021) as a guide.

In regards to the authors addressing this comment from me “The authors could have also considered assessing the quality of the articles selected for the review”, the authors indicated that “The methodological quality of the included studies was assessed using the adapted CASPe guide, where all studies presented a total score between 7 and 9 out of 9.” However, I don’t see this in the manuscript. Could the authors indicate where the manuscript this point “The methodological quality of the included studies was assessed using the adapted CASPe guide, where all studies presented a total score between 7 and 9 out of 9”, has been highlighted?

Author Response

Reviewer 1

We would kindly thank you for the opportunity to improve this work. Please find below the answer to your suggestions.

  • In line 19 the typing error nf has been deleted
  • In line 37 the reference 3 has been removed
  • In line 133 the sentence “The methodological quality of the included studies was assessed using the adapted CASPe guide, where all studies presented a total score between 7 and 9 out of 9” has been introduced
  • In line 141 the grouped word has been included
  • As the reviewer 1 suggested Figure 1 has been modified using Page et al (2021) as a guide. Also, the figure 1 has been changed and the Figure 1 title changed to: Flow diagram showing the search, identification, and selection process for the reviewed articles and the reports excluded by Reason 1: Do not discuss about polyphenols’ action and Gestational Diabetes Mellitus; Reason 2: Do not discuss about Gestational Diabetes Mellitus in relation with Neurodevelopmental and foetus; Reason 3: Reviews. Based on Page et al (2021) [25].
